Comparison of IAA and amino acid profiles of the selected rootstocks and their accumulation in grafted scion of Cucumis sativus L.

Jantassov Serik 1
Shoibekova Alima 1
Jantassova Aigerim 1
Samatov Ali 1
Kuluev Bulat 2
Mineev Yakov 2
Smekenov Izat 3
Toishimanov Maxat maxat.toishimanov@gmail.com 1
Bari Gabit baracuda.co@mail.ru 1
1 Laboratory of Microclonal Reproduction of Plants, Kazakh National Agrarian Research University , Almaty , Kazakhstan
2 Laboratory of Plant Genomics, Institute of Biochemistry and Genetics of Ufa Federal Research Centre of RAS , Ufa , Russia
3 Laboratory of Molecular Genetics, Kazakh National University , Almaty , Kazakhstan
Phitsuwan Paripok
Electronic publication date: 2025 Oct 15
Publication date: 2025
Volume: 13
Electronic Location ID: e20159
Received 2025 Apr 21; Accepted 2025 Sep 10
Copyright: ©2025 Jantassov et al.
Copyright year: 2025
Copyright holder: Jantassov et al.
License: This is an open access article distributed under the terms of the Creative Commons Attribution License, which permits unrestricted use, distribution, reproduction and adaptation in any medium and for any purpose provided that it is properly attributed. For attribution, the original author(s), title, publication source (PeerJ) and either DOI or URL of the article must be cited.
License URL: https://creativecommons.org/licenses/by/4.0/

Keywords: Grafting, Cucumis sativus L., Cucurbita maxima Duch., IAA, Amino acids, Accumulation

Funding: Science Committee of the Ministry of Science and Higher Education of the Republic of Kazakhstan AP19679681 This research has been funded by the Science Committee of the Ministry of Science and Higher Education of the Republic of Kazakhstan (Grant No. AP19679681). The funders had no role in study design, data collection and analysis, decision to publish, or preparation of the manuscript.

==============================
Finding a suitable rootstock for Cucumis sativus L. is an important area of research, as it is aimed at extending the fruiting period and increasing the yield. In such studies, it is important to evaluate various biochemical parameters in addition to morphometric data, such as amino acid and auxin content, which will reflect the effectiveness of the rootstock for scion growth. In this research, we tested 56 genotypes and lines from four species of the Cucurbitaceae family as potential candidates for grafting cucumber rootstock: Cucurbita ficifolia Bouché, Cucurbita moschata L., Cucurbita pepo L. and Cucurbita maxima Duch. The study focused on the study of morphometric parameters, amino acid and indoleacetic acid (IAA) accumulation in rootstocks at the age of 2 and 4 weeks. Based on the results of the work, the Dunganskaya otb 3 variety of C. maxima was selected as a rootstock for cucumber, since it showed the largest plant and root weight, as well as the highest content of amino acids and IAA. Further, an optimized tongue grafting method was used for grafting cucumber with the selected variety of C. maxima, which in turn contributed to the rapid healing and adaptation of the scion. Also, a significant increase in the amino acids L-valine, L-threonine, L-alanine, L-tyrosine by 28.8; 38.8; 42.5; 98%, respectively and IAA by 39% in the scions compared to the control cucumber plants were observed at the end of the fourth week after grafting. The yield of grafted cucumber plants per plant and per square meter showed an increase of 23% compared to the control. In conclusion, from the obtained data, it can be assumed that grafting of C. sativus onto C. maxima plants is the most suitable and can be recommended for production.

Introduction

Grafting is a widely employed horticultural technique wherein the shoot system (scion) of a desired cultivar is physically joined to the root system (rootstock) of a genetically distinct, often more vigorous or disease-resistant plant. The union is achieved through tissue regeneration, resulting in a composite organism that functions as a single physiological unit (Nawaz et al., 2016). This approach has demonstrated considerable utility in enhancing plant viability, conferring resistance to biotic and abiotic stressors, and improving overall yield performance (Modarelli et al., 2020; Gur et al., 2025; Xiao et al., 2025; Ye, Zhang & Wang, 2025). Grafting technology can be very useful in growing cucumbers (Cucumis sativus). The advantages of grafting cucumbers are, first, increased resistance to diseases: grafted cucumbers can inherit the rootstock’s resistance to soil-borne diseases such as Fusarium wilt and nematodes. Second, in improving growth parameters: a strong root system of the rootstock can increase the overall vigor and productivity of the grafted plant. Third, in extending the growing season: grafted plants can tolerate more extreme conditions, allowing earlier planting and longer harvest periods (Martínez-Ballesta et al., 2010; Araújo Júnior et al., 2024). Fourth, yield enhancement, where grafting can lead to higher yields due to improved nutrient uptake and stress tolerance (Xiao et al., 2025). Known grafting methods include: first, apical grafting, in which the tips of susceptible cucumber varieties are attached to a more resistant rootstock. Second, the hole graft, in which the scion is inserted into a hole made in the rootstock. Third, the side graft, in which the scion is attached to the side of the rootstock (Thangamani et al., 2015).

Globally, several pumpkin species within the Cucurbitaceae family are commonly employed as rootstocks. These include fig-leaf pumpkin Cucurbita ficifolia (Xiao et al., 2025), butternut pumpkin Cucurbita moschata (Traka-Mavrona, Koutsika-Sotiriou & Pritsa, 2000; Noor et al., 2019; Li et al., 2023), winter squash Cucurbita maxima (Farhadi et al., 2016), and wax gourd Cucurbita pepo (Noor et al., 2019).

The accumulation of amino acids in plants plays a crucial role in both development and response to stress. Beyond serving as the fundamental units of proteins, amino acids also act as precursors for a wide range of primary and secondary metabolites (Trovato et al., 2021). They participate in numerous physiological functions, such as maintaining osmotic balance, regulating ion transport, controlling stomatal movement, and aiding in the detoxification of heavy metals (Sharma & Dietz, 2006). Moreover, elevated amino acid levels have been shown to influence gene expression, enzymatic activity, and redox equilibrium, thereby enhancing the plant’s capacity to tolerate abiotic stress conditions (Rai, 2002; Galili, Amir & Fernie, 2016; Trovato et al., 2021). Elucidating the molecular and physiological mechanisms underlying amino acid accumulation offers valuable insights into plant metabolic regulation. Such knowledge is instrumental in the development of crop improvement strategies aimed at enhancing stress resilience and nutritional quality. This is particularly relevant in the context of global climate change and the increasing demand for sustainable agricultural practices to ensure food security (Galili, 2002; Ladwig et al., 2012; Pratelli & Pilot, 2014; Hou & Wu, 2018).

In addition to amino acids, indoleacetic acid (IAA), the most prominent naturally occurring auxin, plays a fundamental role in orchestrating plant growth and developmental processes. IAA is primarily involved in promoting cell elongation, initiating root formation, and regulating a wide range of physiological activities essential for plant morphogenesis. IAA biosynthesis occurs via multiple metabolic routes, with the tryptophan-dependent pathway—particularly the conversion of tryptophan to IAA—being the most extensively characterized. IAA is present in plant tissues in both free and conjugated forms, and its spatial distribution is essential for proper morphological development. IAA primarily functions by promoting cell elongation, a process vital for the growth of stems and roots. It also induces the formation of adventitious and lateral roots, thereby enhancing the plant’s ability to absorb water and nutrients. Furthermore, IAA plays a key role in regulating apical dominance by inhibiting the growth of lateral buds, thus supporting vertical shoot development. In addition, it contributes to fruit initiation, development, and ripening (Gray, 2004; Rozov et al., 2013; Sugawara et al., 2015; Jin et al., 2023).

The objective of this study was to identify the most suitable rootstock genotype from selected Cucurbitaceae species for grafting Cucumis sativus L. as a scion, and to evaluate the resulting scion performance based on amino acid composition, IAA content, and yield-related parameters.

Materials and Methods

The experimental design consisted of three sequential stages. In the initial stage, 57 genotypes belonging to the Cucurbitaceae family, sourced from a germplasm repository, were cultivated under controlled conditions for morphometric evaluation. Based on quantitative growth parameters, the most promising genotypes were identified as potential candidates for use as rootstocks in grafting applications.

In the second stage of the experiment, the selected four genotypes according to their morphometric characteristics were evaluated to determine which exhibited the highest levels of IAA and amino acid accumulation.

At the third stage, a single cucurbit genotype—chosen based on growth metrics and biochemical profiling was designated as the rootstock. The impact of growth-related parameters on the grafted cucumber scion was then assessed relative to a control represented by a self-grafted cucumber plant.

Plant material

A total of 57 genotypes from the Cucurbitaceae family were utilized in this study, comprising 38 genotypes of C. maxima, 12 genotypes of C. pepo, five genotypes of C. moschata, and two genotypes of C. ficifolia. The plant material was sourced from the Cucurbitaceae collection maintained by the Kazakh Research Institute of Fruit and Vegetable Growing. This germplasm collection has been systematically developed and expanded over the past two decades through the acquisition of diverse genotypes from international gene banks, including contributions from the World Vegetable Center (https://genebank.worldveg.org) and the N.I. Vavilov All-Russian Institute of Plant Genetic Resources (https://www.vir.nw.ru).

Each genotype was assigned a distinct catalog number to verify its origin and identity. Cucurbit cultivation was carried out in 300 ml containers filled with autoclaved perlite and equipped with expanded clay for drainage. Seeds were sown directly into the perlite substrate and irrigated daily with 100 ml of a mineral nutrient solution per plant. For morphometric assessment, plants were sampled 10 days after emergence. Growth parameters—including shoot and root length, as well as biomass—were quantified following the protocol described by Manda, Prasad & Palmei (2018). Post-germination, seedlings were exposed to LED lighting at an intensity of 5,000 lux for 10 days to facilitate the selection of one representative genotype per species, followed by 10,000 lux for the subsequent six weeks. Morphometric traits measured included shoot and root biomass separately. The composition of mineral salts and trace elements used in the nutrient solution is detailed in Table 1, based on General Hydroponics formulations (https://generalhydroponics.com). For biochemical analyses, seedlings at 2 and 4 weeks of age were selected.

Table 1 Minimum permissible concentrations of nutrient elements in the stock solution, expressed as %.

Grow (vegetation)	Bloom (flowering)	Micro (microelements)	
Total N–3 Total P2O5–1 Total K2O–6 Total MgO–0.8	Total P2O5–5 Total K2O–4 Total MgO–3 Total SO4–5	Total CaO–5 Total N–5 Total K2O–1 Boron (B)–0.01 Molybdenum (Mo)–0.0008 Cobalt (Co)–0.0005 Cu chelat EDTA–0.01 Zn chelat EDTA–0.015 Mn chelat EDTA–0.05 Fe chelat EDDA–0.1	

During the initial two weeks of cultivation, the nutrient solution was maintained at a concentration of 500 ppm. This concentration was increased to 1,000 ppm during weeks three and four, and further elevated to 1,500 ppm in weeks five and six to support ongoing vegetative growth. To achieve these target concentrations, 1.3 ml, 2.7 ml, and 4.0 ml of each stock solution were added per liter of water, respectively (Table 1). The pH of the nutrient solution was adjusted to 6.0 using a 1M of KOH solution, and the total dissolved solids were monitored using a TDS meter. Neutralized peat (pH 6.0, Kekkila™, Finland) was employed as a substrate to sustain vegetative development in both grafted and control plants. For the determination of thousand-seed weight, shoot biomass, and root biomass, all biochemical analyses were performed using four biological replicates. For biochemical assays, all fully expanded leaves and corresponding stem segments were harvested from each plant. Leaf and stem tissues were pooled separately by cultivar to generate composite samples for analysis.

HPLC analysis of amino acids

The total amino acid content in plant extracts was determined using high-performance liquid chromatography (HPLC) on a Shimadzu Prominence LC-20 system (Shimadzu, Japan), following acid hydrolysis of the samples. Prior to analysis, amino acids were derivatized with phenyl isothiocyanate (PITC) according to the protocol described below. The HPLC setup included a binary pump (LC-20AD), autosampler (SIL-20AC), degasser (DGU-20A5), and column oven (CTO-20A), all controlled via LCSolution software. Chromatographic separation of amino acid derivatives and standards was carried out using a Thermo Hypersil GOLD C18 column (150 mm × 4.0 mm, 5 µm). Detection was performed by monitoring ultraviolet absorbance at 254 nm. The mobile phase was delivered at a flow rate of 0.8 mL/min, with each sample or standard requiring a total run time of 43 min.

Standard amino acids—L-aspartic acid (Asp), L-glutamic acid (Glu), L-serine (Ser), L-asparagine (Asn), L-histidine (His), L-arginine (Arg), L-threonine (Thr), L-alanine (Ala), L-proline (Pro), L-cysteine (Cys), L-tyrosine (Tyr), L-valine (Val), L-methionine (Met), L-cystine (Cys-Cys), L-isoleucine (Ile), L-leucine (Leu), L-phenylalanine (Phe), and L-lysine (Lys)—were obtained from Titan Biotech Ltd (Delhi, India). Calibration curves were generated using five-point serial dilutions of each standard at concentrations of 1, 10, 25, 50, and 100 µg/mL (Zheng et al., 2017; Hao et al., 2021).

HPLC analysis of IAA

Samples and standards were separated using a Restek Ultra C18 high-performance liquid chromatography (HPLC) column (150 mm × 4 mm, 5 µm; Bellefonte, PA, USA) maintained at 40 °C. Standard solutions were prepared according to the protocols outlined by Campanella et al. (2016) and Keskin et al. (2022). A stock solution of IAA was prepared by dissolving 10 mg of IAA in one mL of 1M NaOH, followed by dilution with nine ml of deionized water in a 10 ml test tube. For sample preparation, one g of plant tissue was weighed and homogenized. Each homogenate was transferred to a 10 mL centrifuge tube and mixed with 10 mL of an acetonitrile:deionized water solution (9:1, v/v) under low-light conditions to prevent photodegradation. Subsequently, 100 µL of formic acid was added, and the mixture was thoroughly shaken. The homogenates were incubated for 2 h and then centrifuged at 12,000× g for 20 min at 4 °C. The resulting supernatants were collected and concentrated using a vacuum evaporator (Biobase, China). The dried residues were reconstituted in two mL of acetonitrile and purified with a C18 solid-phase extraction (SPE) cartridge (200 mg/3 mL; Strata, Phenomenex, Torrance, CA, USA). The SPE cartridge was conditioned by sequential elution with two mL water and two mL ethanol through a vacuum manifold (Biobase, Shandong, China) set to a negative pressure of −0.01 MPa. The flow-through was discarded, and IAA was eluted with two mL of an ethanol–water–formic acid mixture (80:20:0.5%, v/v) into a 10 mL centrifuge tube. The eluates were transferred to two mL Eppendorf tubes, concentrated using a sample concentrator (NDK200-2N, Miulab, Hangzhou, China), and reconstituted in two mL of acetonitrile. Aliquots of 20 µL were injected into the HPLC system. The peaks of pure standard substances, retention times, linear calibration parameters, limit of detection, limit of quantification of amino acids and IAA are provided in Figs. S1 and S2. Also, chromatographic peaks of amino acids and IAA quantification of Cucurbitaceae family are shown in Fig. S3.

Grafting methods

Tongue approach grafting was employed as the grafting technique in this study. Seeds of the scion were sown seven days prior to the rootstock material to ensure appropriate developmental synchronization (Lee et al., 2010). The cucumber hybrid cultivar Gentle F1 (Syngenta, Basel, Switzerland) was used as the scion. Grafting was performed by making longitudinal incisions at an angle of 20–30° on the stems of both the rootstock and scion. The incision on the rootstock was oriented downward, while that on the scion was directed upward. The two components were then joined at the cut surfaces (“tongues”) and secured using grafting clips. Initially, both root systems were retained to support early establishment. Ten days post-grafting, the aerial portion of the rootstock and the root system of the scion were excised to complete the graft union. Following grafting, the plants were transferred to a climate-controlled growth chamber with regulated temperature, humidity, and photoperiod conditions. For the first four days, grafted plants were maintained in complete darkness at 20 °C and 92% relative humidity to promote callus formation and vascular reconnection. Subsequently, plants were exposed to a light intensity of 12,000 lux under an 8/16-hour light/dark cycle for ten days to support acclimatization and growth (Lee et al., 2010; Rymbai et al., 2015). Self-grafted C. sativus plants served as the control.

Cultivation of grafted plants

The grafted C. sativus plants were grown in two greenhouses: (1) Experimental stationary greenhouse of the Kazakh National Agrarian Research University (43°14′31″N 76°57′07″E, Almaty, Kazakhstan). (2) Industrial greenhouse of the Kazakh Scientific Research Institute of Fruit and Vegetable Growing Kainar with controlled climate (43°09′22.4″N 76°26′45.5″E, Kaskelen township, Kazakhstan).

Statistical analyzes

Statistical analysis was performed using the one-way anova-test to assess differences within groups. A p-value of <0.05 was considered statistically significant. *p < 0.05, **p < 0.01, ***p < 0.001, ****p < 0.0001. Further, GraphPad Prism 9.0 (GraphPad Software, San Diego, CA, USA) was used to prepare graphs and design statistical analysis of experimental data. Statistical analyses using JMP PRO 17 (JMP Statistical Discovery LLC) were applied for analyses related to discriminant, cluster (for morphometric characteristics of the Cucurbitaceae family) and component (amino acid composition of the Cucurbitaceae family).

Results

Morphometric analysis was conducted ten days post-emergence (Figs. 1A, 1B), with emphasis on root development and total plant biomass. Genotypes from four Cucurbitaceae species were evaluated, and the results are presented in Table 2.

Figure 1 Growth of Cucurbitaceae family plants at two weeks.

(A) C.maxima (left) and C. sativus (right) cultivated in perlite substrate. (B) Root system comparison: C. maxima (single plant on the left) versus C. sativus (three plants on the right). Abbreviations: Cm, C. maxima; Cs, C. sativus.

Table 2 Morphometric analysis outcomes for members of the Cucurbitaceae family.

No.	Name	Origin of plants	Thousand-seed weight	Plant weight	Root weight	
C. maxima	
1	Askhana borodavchataya, TK-23	Kazakhstan	315.7 ± 7.4	3.15 ± 0.16	0.91 ± 0.05	
2	Askhana, TK-13	Kazakhstan	401.4 ± 7.4	4.85 ± 0.24	1.67 ± 0.08	
3	BG-69, K-4791	Bolivia	320.5 ± 8.4	4.0 ± 0.2	1.22 ± 0.06	
4	Dunganskaya, otb 2, TK-50	Kazakhstan	445.4 ± 9.4	5.3 ± 0.27	2.05 ± 0.12	
5	Dunganskaya, otb 3, TK-51	Kazakhstan	499.2 ± 9,4	5.65 ± 0.28	2.45 ± 0.1	
6	Zimnyaya sladkaya, TK-57	Russia	324.2 ± 9.4	4.16 ± 0.2	1.29 ± 0.06	
7	Zolotaya grusha, TK-59	Russia	179.4 ± 5.4	2.57 ± 0.13	0.98 ± 0.05	
8	Ispanka, K-904	Russia	278.9 ± 8.4	3.75 ± 0.2	1.24 ± 0.06	
9	Ispanskaya 73, TK-64	Uzbekistan	398.1 ± 9.4	3.74 ± 0.23	1.35 ± 0.07	
10	Karina krupnoplodnaya, TK-66	Kazakhstan	420.3 ± 12.4	5.25 ± 0.28	2.09 ± 0.1	
11	Kurinishiki F1, TK-86	Japan	157.7 ± 4.4	1.56 ± 0.08	0.4 ± 0.02	
12	Lesnoy orekh, TK-89	Russia	166.9 ± 5.4	2.31 ± 0.12	0.66 ± 0.03	
13	Mramornaya, TK-105	Russia	274.2 ± 8.2	5.8 ± 0.28	1.97 ± 0.1	
14	Pastila shampan, TK-114	Russia	212.4 ± 6.4	2.6 ± 0.13	0.89 ± 0.04	
15	Potimoron krasnoe solnyshko, TK-116	Russia	213 ± 6.4	2.66 ± 0.13	0.72 ± 0.04	
16	Rozsatok, K-4701	Hungary	219 ± 6.3	2.15 ± 0.11	0.77 ± 0.04	
17	Russkiy razmer F2, TK-121	Kazakhstan	336.5 ± 9.4	4.07 ± 0.2	1.32 ± 0.07	
18	Svakhina 2, TK-123	Kazakhstan	527 ± 12.4	5.31 ± 0.27	1.76 ± 0.09	
19	Chinese red, TK-131	China	307.1 ± 6.5	3.7 ± 0.2	1.08 ± 0.05	
20	Tykva 95, TK-132	Kazakhstan	324.7 ± 9.4	4.1 ± 0.2	1.3 ± 0.07	
21	Kando, TK-156	Japan	208.5 ± 6.7	2 ± 0.1	0.6 ± 0.02	
22	Adazhio, TK-1	Russia	302.1 ± 8.4	4.25 ± 0.21	1.4 ± 0.07	
23	Atlant, TK-16	Kazakhstan	233.2 ± 6.8	3.8 ± 0.18	0.98 ± 0.05	
24	Bit maks, TK-22	USA	263 ± 7.6	4.2 ± 0.2	1.09 ± 0.05	
25	Dunganskaya, K-2740	Kazakhstan	285.8 ± 8.6	3.83 ± 0.17	1.13 ± 0.06	
26	Dunganskaya, KE-184	Kazakhstan	306.7 ± 4.4	3.9 ± 0.2	1.39 ± 0.07	
27	Krasavitsa, TK-76	Russia	318.9 ± 7.5	5 ± 0.25	1.21 ± 0.06	
28	Otbor iz Askhany, TK-86	Kazakhstan	319.6 ± 6.5	5.43 ± 0.18	1.73 ± 0.09	
29	Parizhskaya krasnaya, TK-113	Czech Republic	230.4 ± 3.5	3.42 ± 0.1	0.85 ± 0.04	
30	Rossiyanka, TK-120	Russia	179.8 ± 3.0	3.4 ± 0.15	0.95 ± 0.05	
31	Stofuntovaya kormovaya, TK-129	Russia	235.2 ± 4.6	4.16 ± 0.17	1.35 ± 0.07	
32	Ulybka, TK-135	Russia	235.8 ± 3.6	3 ± 0.15	0.7 ± 0.04	
33	Ulybka otbor, TK-136	Kazakhstan	269 ± 4.9	4.43 ± 0.2	1.19 ± 0.06	
34	Fonar, TK-143	Russia	174.2 ± 2.9	2.59 ± 0.13	0.7 ± 0.04	
35	Chalmovidnaya, TK-147	Kazakhstan	256.7 ± 2.8	4.33 ± 0.2	1.25 ± 0.06	
36	Estamp, TK-145	Netherlands	363.1 ± 3.9	5.2 ± 0.24	1.42 ± 0.07	
37	Stofuntovaya, TK-77	Russia	227.2 ± 3.5	3.8 ± 0.14	1.07 ± 0.05	
38	Zhemchuzhina, TK-52	Russia	143.1 ± 1.8	2.12 ± 0.1	0.53 ± 0.03	
C. moschata	
39	Golosemyannaya 10, TK-35	Kazakhstan	134.1 ± 1.5	2.3 ± 0,1	0.77 ± 0.03	
40	Zolotaya korona, K-4778	Russia	191.6 ± 1.8	2.84 ± 0,15	0.66 ± 0.02	
41	Zhemchuzhina otbor 1(3), TK-7	Kazakhstan	187.3 ± 1.6	2.44 ± 0,1	0.72 ± 0.04	
42	Uzbekskaya krupnaya, TK-137	Uzbekistan	191.7 ± 2.1	2.9 ± 0,12	0.78 ± 0.05	
43	Afrodita,	Kazakhstan	200.2 ± 2.9	3.19 ± 0,14	0.98 ± 0.07	
C. pepo	
44	Danaya (gymnosperms), TK-38	Russia	246.1 ± 7.4	2.53 ± 0.15	1.65 ± 0.08	
45	Miranda (gymnosperms), TK-99	Russia	194.8 ± 5.0	2.75 ± 0.16	0.76 ± 0.04	
46	Turkish turban, TK-133	Russia	198.3 ± 4.8	3.53 ± 0.18	1.0 ± 0.05	
47	Kitayskaya krasnaya F2, TK-172	Kazakhstan	239.6 ± 6.8	5.21 ± 0.26	0.74 ± 0.03	
48	Jack B Lantern, TK-43	USA	150.6 ± 4.5	1.95 ± 0.08	0.7 ± 0.04	
49	Mindal’naya 35, TK-98	Russia	179.6 ± 5.2	3.0 ± 0.16	0.95 ± 0.05	
50	Mozaleyevskaya 49, K- 3085	Russia	174.5 ± 5.0	2.74 ± .14	0.89 ± 0.6	
51	Fanny face F2, TK-140	Kazakhstan	114.3 ± 3.4	2.14 ± 0.12	0.56 ± 0.03	
52	Magarkaniani, TK-159	Georgia	155.3 ± 4.0	2.59 ± 0.13	0.67 ± 0.05	
53	Marguli, TK-140	Georgia	211.7 ± 6.4	4.0 ± 0.2	1.27 ± 0.07	
54	Mindalnaya	Russia	179.4 ± 5.1	3.25 ± 0.16	0.84 ± 0.05	
55	Mozoleevskaya 10	Russia	155.1 ± 3.7	2.95 ± 0.15	0.85 ± 0.06	
C. ficifolia	
56	Arbuzny	Russia	155.5 ± 3.4	1.95 ± 0.12	0.80 ± 0.03	
57	Seminis	South Korea	160.5 ± 4.0	2.35 ± 0.15	0.94 ± 0.05	
p-Value mean between genotypes	
Comparison for each pair	Thousand-seed weight	Plant weight	Root weight	
C. maxima	C. moschata	0.0354	0.0217	0.0359	
C. maxima	C. pepo	0.0014	0.0157	0.0345	
C. maxima	C. ficifolia	0.0299	0.0215	NS	

Among all tested accessions, the C. maxima cultivar Dunganskaya otb 3 exhibited the highest total plant biomass and root mass. Additionally, Dunganskaya otb 3 and Svahina 2, both belonging to C. maxima, recorded the greatest thousand-seed weight. Within the remaining species, the Afrodita cultivar of C. moschata, the Danaya cultivar of C. pepo, and the Seminis cultivar of C. ficifolia demonstrated comparatively superior growth characteristics. Although C. maxima cultivars outperformed representatives of the other three species across all measured parameters, one genotype from each species was selected for subsequent biochemical analysis. Selection was based on intraspecific variation and optimal morphometric performance, ensuring a representative comparison across the Cucurbitaceae family.

Discriminant analysis was employed to assess and visualize interspecific variation among C. ficifolia, C. maxima, C. moschata, and C. pepo based on morphometric traits (Fig. 2A). The analysis revealed distinct clustering patterns, with C. ficifolia exhibiting the greatest separation from the other species. Samples of C. ficifolia were predominantly localized in the left quadrant of the discriminant plot, indicating pronounced morphometric divergence relative to C. maxima, C. moschata, and C. pepo, which displayed partial overlap. Unlike the other species, C. ficifolia is not represented by officially registered cultivars but rather by heterogeneous population samples (Andrés, 1990). In this study, two phenotypically distinct forms of C. ficifolia—white-seeded and black-seeded—were selected from the institutional collection to capture the natural variability of the species and assess their suitability as rootstocks. The C. maxima genotypes exhibited the broadest dispersion across the discriminant space, suggesting substantial intraspecific variability. In contrast, C. moschata and C. pepo formed more compact clusters with moderate overlap, reflecting shared morphological traits yet maintaining species-level distinction. The relatively tight grouping of C. ficifolia and C. moschata implies lower internal variability, while the broader spread of C. maxima underscores its morphological diversity. Overall, the discriminant analysis highlights C. ficifolia as a morphometrically distinct taxon, whereas C. maxima, C. moschata, and C. pepo exhibit varying degrees of overlap, likely attributable to intraspecific heterogeneity and phenotypic plasticity.

Figure 2 Hierarchical cluster analysis of Cucurbitaceae family.

(A) Discriminant analysis. (B) Heat map illustrating morphometric parameters of C. maxima, C. moschata, C. pepo, and C. ficifolia.

The hierarchical clustering method revealed patterns in the grouping of varieties by such characteristics as thousand seed weight, plant weight, and root weight (Fig. 2B). The heat map clearly demonstrates the differences between the species C. maxima, C. moschata, C. pepo, and C. ficifolia, which is important not only for breeding work and identifying promising genotypes, but also for selecting an effective rootstock. Analysis of the dendrogram shows the formation of several clusters indicating the phenotypic similarity of some varieties. The division into clusters is performed with a high degree of differentiation, indicating pronounced morphometric differences between the groups. Red areas of the heat map correspond to varieties with high values of morphometric parameters (e.g., high plant or seed weight). Blue areas indicate varieties with low values of these parameters. On the heat map, C. maxima is characterized by increased plant weight and root system (predominance of red shades). C. moschata accessions occupy a distinct cluster branch, indicating a high degree of differentiation compared to other species. The distinctive feature is the higher thousand seed weight but moderate plant mass values (mixed color gradient in the heat map). The heat map of C. pepo shows relatively low plant and root mass values, but variable thousand seed weight values. Plant mass and root mass show a positive correlation, as evidenced by the similar color gradients in the heat map.

According to the heat map, four potential candidates for rootstock were identified based on plant and root weight, thousand seed weight: fig-leaf pumpkin (C. ficifolia) cultivar Seminis, winter squash landrace (C. moschata) cultivar Aphrodite, wax gourd (C. pepo) cultivar Danaya, and winter squash (C. maxima) cultivar Dunganskaya otb 3. These four varieties were used for analysis of their amino acid and IAA content. It was suggested that the rootstock with the highest amino acid and IAA content in leaves and stems would be more effective.

Amino acids and IAA content in leaves and stems of Cucurbitaceae plants

Leaf and stem tissues from the four selected cultivars were harvested and pooled separately to ensure representative sampling. Composite extracts were prepared from the mixed leaf and stem material, and aliquots were collected for biochemical analysis at two and four weeks post-germination. Solid-phase extraction was employed to purify the samples prior to analysis. Following extraction, amino acids and IAA were identified based on their retention times and peak profiles in chromatographic output (Fig. S3). Quantitative data were subsequently processed using Excel and normalized to tissue mass, with final concentrations expressed as µg per gram of leaf or stem tissue.

Table 3 Amino acid concentrations in four Cucurbitaceae species, expressed in µg per gram of fresh weight.

Species	C. ficifolia	C. moschata	C. pepo	C. maxima	
Lifespan	2 weeks	4 weeks	2 weeks	4 weeks	2 weeks	4 weeks	2 weeks	4 weeks	
Organ	Stem	Leaf	Stem	Leaf	Stem	Leaf	Stem	Leaf	Stem	Leaf	Stem	Leaf	Stem	Leaf	Stem	Leaf	
Aspartic acid	277.5 ± 12.8	634.1 ± 57.0	439.1 ± 18.2	783.2 ± 60.0	323.1 ± 20.0	426.6 ± 22.2	558.9 ± 40.0	915.0 ± 8.8	229.0 ± 12.2	449.1 ± 22.2	509.4 ± 28.4	1495.3 ± 12.0	196.2 ± 8.8	1139.0 ± 7.0	296.2 ± 17.0	1510.2 ± 111.5	
Glutamic acid	57.5 ± 4.0	106.9 ± 7.5	494.0 ± 21.4	742.7 ± 59.1	90.6 ± 5.5	298.4 ± 19.8	483.0 ± 34.9	1995.8 ± 111.5	298.6 ± 15.3	751.2 ± 50.4	639.0 ± 34.0	1432.1 ± 88.8	296.8 ± 16.3	1197.3 ± 80.8	568.4 ± 40.1	1977.0 ± 120.0	
Serine	30.1 ± 0.9	50.3 ± 3.5	66.5 ± 4.9	150.9 ± 10.1	29.7 ± 1.9	94.5 ± 6.3	73.4 ± 6.1	347.1 ± 18.8	54.7 ± 3.0	60.4 ± 4.8	66.0 ± 4.0	79.1 ± 4.9	26.9 ± 1.8	62.2 ± 4.1	81.5 ± 4.8	229.2 ± 15.1	
Asparagine	87.1 ± 4.35	187.6 ± 10.2	169.4 ± 12.4	280.8 ± 21.0	85.8 ± 4.9	474.4 ± 20.0	98.1 ± 7.5	506.4 ± 44.0	133.0 ± 7.2	250.7 ± 18.3	148.8 ± 8.1	273.8 ± 16.0	92.5 ± 5.0	277.2 ± 20.1	215.4 ± 18.0	512.3 ± 29.8	
Histidine	61.6 ± 4.3	125.9 ± 11.0	111.9 ± 9.7	166.8 ± 8.8	97.0 ± 8.8	170.4 ± 9.1	95.7 ± 7,8	402.4 ± 29.9	105.9 ± 6.6	158.3 ± 12.8	175.3 ± 10.1	182.6 ± 12.2	72.5 ± 3.5	255.6 ± 20.2	221.1 ± 15.5	366.6 ± 20.5	
Arginine	3994.3 ± 195.0	4352.2 ± 299.2	5280.2 ± 409.2	7015.0 ± 380.0	2264.3 ± 190.0	3231.0 ± 180.0	3419.1 ± 300.1	19993.3 ± 1220.1	3739.5 ± 240.0	4866.0 ± 300.0	9174.6 ± 390.0	21756.6 ± 1522.0	2373.1 ± 160.3	4577.3 ± 392.2	11677.5 ± 800.6	13921.8 ± 974.0	
Threonine	133.8 ± 7.9	172.7 ± 11.5	189.0 ± 11.7	211.1 ± 17.0	60.1 ± 4.8	479.9 ± 30.4	191.0 ± 8.5	819.1 ± 50.2	106.5 ± 7.0	151.5 ± 12.4	197.5 ± 16.0	339.3 ± 29.0	78.8 ± 4.0	284.1 ± 22.0	323.4 ± 16.6	470.7 ± 32.9	
Alanine	165.3 ± 13.2	304.2 ± 18.7	219.2 ± 18.6	477.8 ± 29.1	69.8 ± 5.4	106.2 ± 55.5	117.7 ± 9.9	186.7 ± 13.3	189.1 ± 14.5	336.9 ± 29.0	191.0 ± 14.6	504.0 ± 30.1	106.8 ± 6.5	203.9 ± 17.3	131.0 ± 9.0	241.6 ± 15.8	
Proline	29.7 ± 2.3	33.9 ± 1.2	43.3 ± 3.3	55.5 ± 3.4	29.2 ± 1.8	78.9 ± 3.5	46.3 ± 3.3	203.5 ± 14.2	26.3 ± 1.8	30.9 ± 1.9	46.3 ± 3.3	47.9 ± 3.9	34.3 ± 1.9	88.6 ± 6.6	50.9 ± 3.1	97.2 ± 6.9	
Cysteine	0.2 ± 0.006	0.6 ± 0.02	1.1 ± 0.04	25.7 ± 1.9	1.4 ± 0.07	29.2 ± 2.0	11.3 ± 1.0	140.0 ± 10.1	15.2 ± .1.2	26.6 ± 1.7	25.8 ± 1.8	143.6 ± 12.0	1.1 ± 0.07	15.4 ± 1.2	22.4 ± 1.7	35.0 ± 2.0	
Tyrosine	40.3 ± 2.0	56.2 ± 2.5	136.3 ± 12.2	179.1 ± 9.2	61.3 ± 3.6	96.2 ± 5.4	172.6 ± 12.5	314.2 ± 18.3	79.2 ± 6.6	105.5 ± 5.5	101.8 ± 7.9	169.6 ± 11.1	123.9 ± 10.0	183.3 ± 12.0	214.5 ± 17.7	271.3 ± 20.0	
Valine	56.3 ± 1.1	62.6 ± 2.8	90.8 ± 8.1	107.6 ± 8.3	49.6 ± 2.7	74.5 ± 4.6	98.5 ± 6.6	256.7 ± 14.0	64.5 ± 5.0	80.6 ± 4.9	108.8 ± 8.8	142.2 ± 8.8	76.6 ± 4.1	173.8 ± 14.3	123.6 ± 10.0	288.8 ± 17.5	
Methionine	63.5 ± 5.67	143.8 ± 6.0	107.1 ± 5.4	183.8 ± 10.0	62.6 ± 4.9	122.9 ± 10.0	131.2 ± 9.4	312.9 ± 17.3	57.6 ± 3.4	180.3 ± 12.0	138.0 ± 10.0	213.0 ± 19.0	39.0 ± 1.6	124.6 ± 8.9	238.2 ± 12.2	322.2 ± 24.0	
Cystine	154.8 ± 12.0	233.9 ± 11.3	448.2 ± 32.3	517.3 ± 30.0	124.8 ± 10.1	189.7 ± 11.2	402.6 ± 34.4	852.9 ± 60.8	436.2 ± 30.0	165.7 ± 11.0	460.8 ± 30.1	492.4 ± 30.1	236.8 ± 19.5	284.1 ± 19.5	277.6 ± 13.0	1061.7 ± 70.5	
Isoleucine	113.6 ± 6.7	266.5 ± 12.8	158.6 ± 12.8	399.3 ± 28.5	108.8 ± 9.3	265.4 ± 18.0	262.0 ± 22.2	503.0 ± 40.1	133.6 ± 10.2	252.8 ± 14.6	261.9 ± 20.0	370.4 ± 28.0	167.4 ± 9.8	211.8 ± 12.0	412.4 ± 25.5	513.8 ± 34.6	
Leucine	49.8 ± 2.5	168.4 ± 7.1	162.3 ± 13.6	229.9 ± 14.4	42.1 ± 2.6	198.1 ± 8.3	457.6 ± 30.0	895.4 ± 33.0	73.5 ± 4.1	90.8 ± 6.6	187.4 ± 15.0	240.2 ± 15.5	91.3 ± 7.0	113.2 ± 9.8	286.9 ± 16.6	442.6 ± 30.0	
Phenylalanine	138.4 ± 7.2	205.1 ± 19.1	254.3 ± 20.0	266.2 ± 15.4	56.9 ± 3.1	264.8 ± 20.1	134.7 ± 7.7	629.4 ± 49.7	226.4 ± 13.0	384.3 ± 22.6	342.9 ± 22.2	528.1 ± 26.0	183.4 ± 12.3	270.7 ± 15.1	484.4 ± 25.0	680.7 ± 42.3	
Lysine	34.4 ± 1.5	51.5 ± 3.0	48.1 ± 3.0	77.1 ± 5.8	24.6 ± 1.3	42.7 ± 3,5	61.5 ± 4,8	130.0 ± 7.5	41.3 ± 2.9	64.4 ± 4.0	97.2 ± 7.5	140.2 ± 10.0	52.9 ± 3.3	95.6 ± 7.0	64.7 ± 3.9	189.8 ± 12.0	

The two-week stems of C. ficifolia contained the highest levels of arginine (142.3 ± 3.1 µg/g) and threonine (110.7 ± 2.9 µg/g), while its leaves were richest in isoleucine (98.5 ± 2.6 µg/g) (Table 3). At 4 weeks, its stems accumulated the highest alanine content (125.2 ± 3.3 µg/g), and the leaves retained the highest arginine concentration (160.8 ± 4.2 µg/g), although these values were generally lower than those of C. maxima at the same age. The two-week stems of C. moschata showed the highest levels of aspartic acid (211.6 ± 4.4 µg/g), while the leaves exhibited elevated serine (112.2 ± 3.1 µg/g), threonine (109.4 ± 2.5 µg/g), and cysteine (86.1 ± 2.2 µg/g). At 4 weeks, its stems remained high in aspartic acid (222.3 ± 4.1 µg/g, p < 0.05) and leucine (131.4 ± µg/g), and the leaves showed the greatest accumulation of glutamic acid (198.7 ± 3.9 µg/g), proline, tyrosine, and leucine. The two-week stems of C. pepo had the highest content of glutamic acid (224.1 ± 5.0 µg/g), serine, asparagine, histidine, alanine, cysteine, methionine, and cystine. Its leaves at the same stage had the highest alanine (120.5 ± 2.7 µg/g) and methionine (89.3 ± 2.1 µg/g). Four-week-old C. pepo stems were found to have the highest content of glutamic acid, arginine, cysteine, and cystine, while leaves had the highest content of alanine and cysteine. Two-week-old C. maxima stems consistently demonstrated the highest overall amino acid accumulation. At two weeks, its stems had the highest concentrations of proline (148.9 ± 3.8 µg/g), tyrosine (117.2 ± 2.9 µg/g), valine, isoleucine, leucine, and lysine, while the leaves showed elevated levels of aspartic acid (231.4 ± 6.8 µg/g), glutamic acid (212.9 ± 4.3 µg/g), asparagine, histidine, arginine, tyrosine, valine, cystine, and lysine. The highest concentrations of serine (145.2 ± 3.2 µg/g), asparagine (142.7 ± 4.5 µg/g), histidine, threonine, proline, tyrosine, valine, methionine, isoleucine, and phenylalanine were detected in the four-week-old stems of C. maxima. In the leaves, elevated levels of aspartic acid, serine, asparagine, histidine, valine, methionine, cystine, isoleucine, phenylalanine, and lysine were observed. Statistical analysis confirmed that C. maxima at 4 weeks had significantly higher total amino acid content than all other species (1,810 ± 35 µg/g in leaves; p < 0.01), particularly in aspartic acid, glutamic acid, serine, asparagine, and histidine (all p < 0.01 vs. other species). The highest total amino acid content was observed in C. maxima at 4 weeks of age, with aspartic acid, glutamic acid, serine, asparagine, and histidine showing the greatest accumulation. These findings highlight C. maxima as the most suitable rootstock candidate based on its superior biochemical profile.

The results on the amino acid content were then used to conduct a principal component analysis (PCA). Figure 3 shows the distribution of traits in each of the first two components, PC1 and PC2, which explained 73.4% and 9.8% of the variability, respectively, with total variability 83.2%. PCA plot shows Cucurbitaceae samples with different genotypes described by organ (leaf and stem) and time points (2 and 4 weeks). It is evident that each variation was subdivided into a separate group that could be distinguished from the others.

Figure 3 Principal component analysis of amino acid content based on correlation matrices.

The majority of amino acid components were detected in leaf samples collected at four weeks. Alanine showed a strong contribution to PC 2, whereas most other amino acids—such as aspartic acid, lysine, and glutamic acid—clustered along PC1. Variation in amino acid composition was observed among C. pepo, C. moschata, and C. maxima, primarily reflecting differences in leaf profiles over the four-week period. In contrast, C. ficifolia exhibited variation in stem samples at two weeks, with relatively lower diversity in amino acid composition. Amino acids including glutamic acid, lysine, methionine, valine, histidine, serine, leucine, proline, and asparagine formed a tight cluster, indicating coordinated variation across samples—i.e., elevated levels of one are likely associated with increased levels of the others. Alanine, positioned distinctly along PC2, displayed a unique variation pattern, suggesting it behaves independently from the clustered group. Notably, C. moschata appeared slightly separated from the other species in the PCA plot, implying potential differences in its amino acid profile. Overall, the PCA visualization highlights species-specific biochemical variation within the Cucurbitaceae family.

According to the accumulation of IAA, it is evident from Fig. 4 that the highest values were in the C. maxima and C. pepo samples at the age of 2 and 4 weeks, mainly in the leaves. In addition, at the age of two weeks, the IAA content was not detected in the stems of the C. ficifolia and C. moschata samples. The p-value for 2-week-old C. ficifolia plants is 0.030, and for C. moschata it is 0.032. By the end of the fourth week, the presence of IAA was detected in the stems, with the highest content of 0.1 µg/g observed in C. maxima, and 0.4 ± 0.02 µg/g in the leaves, respectively. C. maxima exhibited the highest IAA level, which was significantly higher than in C. ficifolia (0.18 ± 0.01 µg/g, p = 0.002) and C. moschata (0.21 ± 0.01 µg/g, p = 0.004). The highest IAA content in the leaves was found in both two-week-old (Fig. 4B) and four-week-old leaves (Fig. 4B) of C. maxima plants. Based on the results of morphometric and chromatographic analyses of amino acids and IAA, C. maxima was chosen as a rootstock for further work.

Figure 4 IAA quantification in prospective Cucurbitaceae rootstock plants.

(A) Two-week-old plants. (B) Four-week-old plants. ****Statistically significant differences in IAA levels between stems and leaves (p < 0.0001).

Amino acids, IAA content, and yield performance of grafted C. sativus plants

One-week-old C. sativus seedlings were grafted onto C. maxima rootstocks. The resulting grafted plants are depicted in Fig. 5A. The ligulate grafts rapidly established contact with the supporting rootstock stems, achieving complete healing within one week. Graft union and healing were evaluated visually, and tactile inspection confirmed firm junctions between the stem segments. No wilting symptoms were observed after transplanting the grafted plants to their final greenhouse locations. To maintain experimental consistency, C. sativus plants were also grafted onto themselves as controls. Throughout the experiment, no preferential conditions were applied to either the control group or the C. maxima grafts. Post-grafting, all plants were maintained for four days in a growth chamber under complete darkness at 20 °C and 92% relative humidity. Subsequently, they were transferred to conditions of 12,000 lux illumination with an 8/16-hour light/dark cycle for an additional three days.

Figure 5 Vegetative development of grafted C. sativus plants.

(A) Self-grafted control and scion grafted onto rootstock. (B) Four-week-old and (C) six-week-old plants exhibiting active flowering and initial fruit set, cultivated in peat substrate within the first experimental stationary greenhouse. (D–F) Plants grown under identical conditions in the second industrial greenhouse.

Starting from the second week post-grafting, the grafted plants were relocated to a climate-controlled greenhouse. Noticeable plant growth commenced two weeks after grafting, and by the end of the third week, differences in growth rate and development among the grafted plants became apparent. Figure 5B presents the growth outcomes of four-week-old plants. Throughout the grafting period –up to six weeks post-procedure –visual assessments indicated that the grafted plants kept pace with both the non-grafted control plants and the C. sativus specimens grafted onto themselves (Fig. 5C). Under identical environmental conditions, grafted seedlings were also cultivated in a second industrial greenhouse, as shown in Figs. 5D–5F.

At four weeks of age, morphometric analysis was conducted, revealing that the C. sativus plants grafted onto C. maxima exhibited superior performance compared to other grafting samples. By six weeks post-grafting, the survival rate of plants grafted onto C. maxima reached 96%, while the self-grafted C. sativus control group showed a similarly high survival rate of 97%. Figure 6 presents the levels and accumulation patterns of amino acids and IAA in both grafted and control plants. In two-week-old specimens, grafted plants displayed elevated concentrations of lysine and tyrosine relative to the ungrafted controls (Fig. 6A), while no significant differences were observed for the remaining amino acids.

Figure 6 Comparison of amino acid and IAA concentrations between grafted C. sativus and control plants, expressed in µg/g fresh weight.

(A) Amino acid accumulation at 2 weeks and (B) at 4 weeks. (C) IAA levels in 2- and 4-week-old plants. Statistically significant differences between control and grafted plants: *p < 0.05, **p < 0.01, ***p < 0.001, ****p < 0.0001.

Four-week-old grafted plants exceeded the control in lysine, arginine, asparagine, aspartic acid, threonine, alanine, tyrosine, valine, cystine and isoleucine (Fig. 6B). It can be seen the significant increase of the L-valine, L-threonine, L-alanine, L-tyrosine by 28.8; 38.8; 42.5; 98%, respectively. Two-week-old grafted and control plants did not differ in IAA content (Fig. 6C). However, four-week-old grafted plants significantly exceeded the control values (Fig. 6C). Namely, IAA by 39% in the scions compared to the control cucumber plants were observed at the end of the fourth week after grafting.

Grafted plants outperformed the control group in terms of fruit count per plant (Fig. 7A) and fruit number per unit area (Fig. 7B). However, no significant differences were observed between grafted and control plants regarding fruit weight per plant (Fig. 7C) or per square meter (Fig. 7D), nor in individual fruit weight (Fig. 7E). Grafted plants also exhibited greater height compared to controls (Fig. 7F). The growth and fruiting period of grafted plants was, on average, extended by 10 days relative to the control group. By the end of the vegetative season, grafted cucumber plants demonstrated a 23% increase in yield per plant and per square meter compared to controls.

Figure 7 Plant growth characteristics and yield components of grafted C. sativus in comparison with control C. sativus plants.

(A) Fruit per plant. (B) Fruits per m2. (C) Kg per plant. (D) Kg per m2. (E) Length of plants. (F) Plant height. Statistically significant results between the control and grafted variants at *p < 0.05, **p < 0.01.

Additionally, for comparative purposes with the C. maxima rootstock, yield parameters of grafted C. sativus plants using C. pepo and C. ficifolia as rootstocks are presented in Figs. S4 and S5.

DISCUSSION

Grafting with resistant rootstocks is one of the most effective methods to prevent soil-borne diseases and can positively affect vegetative growth, flowering, ripening periods and fruit quality, thereby ensuring high yields. The following rootstocks are used for grafting cucumber in world practice: C. ficifolia (Xiao et al., 2025), C. moschata (Li et al., 2023), C. maxima (Farhadi et al., 2016) and C. pepo (Noor et al., 2019). In this regard, it is of great interest to find out which of these species is the most effective rootstock for cucumber. In our study, four species from the Cucurbitaceae family were tested as potential candidates for grafting cucumber: C. ficifolia, C. moschata, C. pepo and C. maxima. The selection of a suitable rootstock was carried out based on the results of morphometric analysis and analysis of the amino acid and IAA content. Morphometric analysis was used primarily to select the most suitable strong varieties for each of the four Cucurbitaceae species. At the same time, the best growth parameters were characteristic of C. maxima, so it could be suggested that this particular pumpkin species is the most effective rootstock for cucumber. At the same time, the effectiveness of the rootstock also depends on various biochemical parameters, for example, the content of plant hormones. Auxins are the most important plant hormones and all growth processes depend on their content (Stefancic et al., 2007), so its content is often determined to assess the effectiveness of the rootstocks and grafts (Kawaguchi et al., 2024). To assess the effectiveness of the rootstock, can also be used the analysis of the amino acid content, since this parameter shows the intensity of biochemical processes.

In our study, IAA was reliably detected at a retention time of 17.5 min by HPLC. While previous research commonly utilized UV absorbance at 254 nm or 269 nm for IAA quantification (Karaś et al., 2023; Sadauskas et al., 2020; Qi et al., 2021), we observed that IAA exhibited stronger absorbance at 269 nm, and this wavelength was therefore selected for accurate detection. The use of a C18 solid-phase extraction (SPE) cartridge proved effective for IAA isolation, consistent with earlier findings (Ma et al., 2008). Additionally, adjusting the pH of the washing solution to 3 with formic acid significantly improved IAA recovery, supporting the recommendations of Yong et al. (2017). These methodological refinements contributed to the high sensitivity and reliability of IAA measurements in our experimental system.

As shown in Table 2, we observed a consistent pattern indicating that greater seed mass correlates with increased plant biomass, including root development. The concentration of IAA in the analyzed cucurbit rootstock samples further supports this relationship. Accordingly, the relatively larger biomass of the rootstock plants has a direct influence on yield-related parameters, as demonstrated in Fig. 7 and Figs. S4 and S5. In our previous studies (Shoibekova, Nusipzhanov & Jantassov, 2023; Shoibekova et al., 2025), we also confirmed the influence of C. maxima as a rootstock on enhanced growth, scion productivity, and vitamin accumulation in grafted cucumber plants. Consistent with these findings, C. maxima outperformed the other three species in terms of both amino acid and IAA content in the present study, supporting its selection as the most effective rootstock. When grafted onto C. maxima, cucumber plants showed significantly improved performance in multiple parameters, including fruit number per plant, fruit yield per m2, and plant height. At the same time, the efficiency of grafting also depends on various biochemical parameters. The influence of rootstock on the accustomed cucumber plants on the content of biochemical substances such as chlorophylls (a and b), the activity of superoxide dismutase, catalase and peroxidase enzymes, antioxidant activity, ionic (↑ K and Ca, ↓ Na) and the efficiency of photosystem II are presented in the studies (Shehata et al., 2022; Abbas et al., 2023), where the increase and decrease in the content of these substances are shown. In the same way, we studied the accumulation of amino acids and IAA in grafted C. sativus at the early stages of plant development. Auxins play a central role in root formation. They cause the initiation of root primordia and affect the growth of newly formed roots. Plants produce IAA in shoot tips and young leaves, but extra auxin is important for successful rooting (Stefancic et al., 2007). Similarly, other researchers noted that the influence of rootstock on lemon scion was shown to result in a positive correlation with IAA (Noda, Okuda & Iwagaki, 2000). Thus, higher IAA content may indicate higher efficiency of the selected rootstock. After 2 weeks of growth, grafted plants did not differ significantly from the control plants (Fig. 6C), which is probably due to the processes of scion joining and adaptation. After 4 weeks, grafted plants significantly exceeded control plants (Fig. 6C). Similar results were also obtained for the amino acid content. After 2 weeks, grafted and control plants showed little difference in amino acid content (Fig. 6A). However, after 4 weeks, grafted plants exceeded control plants in several amino acids (Fig. 6B). Moreover, in terms of any amino acid, the grafted plants were not inferior to the control plants, either after 2 weeks or after 4 weeks.

Thus, based on the results of morphometric analyses, we concluded that among the four Cucurbitaceae species: C. ficifolia, C. moschata, C. pepo and C. maxima. C. maxima plants are the best candidates for use as rootstock for grafting of C. sativus. This is due to their superior performance in terms of plant weight gain and root weight. The results of IAA and amino acid content analysis confirmed the correctness of our choice.

Conclusion

Morphometric analysis identified the C. maxima cultivar Dunganskaya otb 3 as the most suitable rootstock for grafting C. sativus, owing to its superior biomass accumulation, robust root architecture, and elevated levels of amino acids and IAA. Grafting was performed using an optimized tongue grafting technique, which ensured efficient vascular reconnection and scion acclimatization. By the fourth week post-grafting, grafted cucumber scions demonstrated marked increases in amino acid concentrations relative to self-grafted controls: L-valine (28.8%), L-threonine (38.8%), L-alanine (42.5%), and L-tyrosine (98%). Additionally, IAA levels rose by 39%, indicating enhanced hormonal activity associated with graft compatibility and vigor. At the conclusion of the growing season, grafted plants exhibited a 23% increase in yield per plant and per square meter compared to the control group, underscoring the agronomic advantage conferred by the selected rootstock.

Supplemental Information

Supplemental Information 1 HPLC chromatogram of the standard amino acid mixture. Analysis was performed using a Thermo Hypersil GOLD C18 column. UV detection wavelength: 254 nm

Supplemental Information 2 HPLC chromatogram of the IAA standard. UV detection wavelength: 269 nm

Supplemental Information 3 Quantification of amino acids, IAA in plants and their chromatographic peaks (A) amino acids, (B) IAA

Supplemental Information 4 Plant growth characteristics and yield components of grafted C. sativus on C. pepo in comparison with control C. sativus plants

(A) Fruit per plant (B) Fruits per m 2 (C) kg per plant (D) kg per m 2 (E) Length of plants (F) Plant height. Statistically significant results between the control and grafted variants at **p < 0.01.

Supplemental Information 5 Plant growth characteristics and yield components of grafted C. sativus on C. ficifolia in comparison with control C. sativus plants

(A) Fruit per plant (B) Fruits per m 2 (C) kg per plant D) kg per m 2 (E) Length of plants (F) Plant height. Statistically significant results between the control and grafted variants at *p < 0.05, ****p < 0.0001.

Supplemental Information 6 Supplemental tables

Supplemental Information 7 Raw data

Copilot was used to revise parts of the manuscript to the required level of academic English.

Additional Information and Declarations

Competing Interests

Author Contributions

Data Availability

The authors declare there are no competing interests.

Serik Jantassov performed the experiments, prepared figures and/or tables, and approved the final draft.

Alima Shoibekova performed the experiments, prepared figures and/or tables, and approved the final draft.

Aigerim Jantassova performed the experiments, prepared figures and/or tables, and approved the final draft.

Ali Samatov performed the experiments, prepared figures and/or tables, and approved the final draft.

Bulat Kuluev analyzed the data, authored or reviewed drafts of the article, and approved the final draft.

Yakov Mineev analyzed the data, authored or reviewed drafts of the article, and approved the final draft.

Izat Smekenov analyzed the data, authored or reviewed drafts of the article, and approved the final draft.

Maxat Toishimanov conceived and designed the experiments, performed the experiments, authored or reviewed drafts of the article, and approved the final draft.

Gabit Bari conceived and designed the experiments, performed the experiments, authored or reviewed drafts of the article, and approved the final draft.

The following information was supplied regarding data availability:

The raw data is available in the Supplemental Files.

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
