# Peer review of "Comparison of IAA and amino acid profiles of the selected rootstocks and their accumulation in grafted scion of Cucumis sativus L"

_PeerJ, doi:10.7717/peerj.20159_

## Round 0.1 · original submission · Major Revisions

Please provide a clear rationale for each experimental section and ensure that the statistical analysis is appropriately applied and reported.

·

Basic reporting

The article "Comparison of IAA and amino acids profiles of the selected rootstocks and their accumulation in grafted scion of Cucumis sativus L." is written in clear, unambiguous, and professional English. It is well organized and provides a thorough review of relevant literature, along with sufficient field context. The article’s structure is exemplary, with appropriately included figures, tables, and raw data, ensuring that the results are self-contained and directly support the stated hypothesis. However, the manuscript would be further enhanced by explicitly addressing the current research gap, which would better underscore the study's significance and novelty. Additionally, any grammatical errors should be corrected before final acceptance.

Experimental design

The experimental design represents original primary research that aligns well with the journal’s Aims and Scope. The research questions are clearly defined, relevant, and meaningful, providing a solid framework for the investigation. However, the manuscript would benefit from a more explicit discussion of the knowledge gap it aims to address.

Overall, the study reflects a rigorous investigation that meets high technical and ethical standards. The methods are described in sufficient detail to enable replication, which further strengthens the manuscript.

Additional comments and specific suggestions are provided in the attached PDF.

Validity of the findings

The experimental design represents original primary research that aligns well with the journal’s Aims and Scope. The research questions are clearly defined, relevant, and meaningful, providing a solid framework for the investigation. However, the manuscript would benefit from a more explicit discussion of the knowledge gap it aims to address.

Overall, the study reflects a rigorous investigation that meets high technical and ethical standards. The methods are described in sufficient detail to enable replication, which further strengthens the manuscript.

Additional comments and specific suggestions are provided in the attached PDF.

Reviewer 2 ·

Basic reporting

All figures need to include legends, and some figures require explanations of the control groups.

Experimental design

In the experimental methods section, the selection of materials and the measurement targets were not clearly described.

Validity of the findings

Some of the results is speculative and should avoid exaggeration or subjective interpretation of the results.

Additional comments

“The article titled ‘Comparison of IAA and amino acids profiles of the selected rootstocks and their accumulation in grafted scion of Cucumis sativus L.’ conducted both grafting and seedlings of Cucumis species, and measured the contents of IAA and amino acids. The study found that the yield of grafted cucumber plants increased by 23% per plant and per square meter. However, the article still has some logical issues and requires major revision and careful editing.

1.The specific origin of the 57 genotypes of Cucurbitaceae should be clearly stated, not just the country. In addition, how many replicates were conducted for the measurements of thousand-seed weight, plant weight, and root weight? Statistical analysis such as ANOVA is needed to compare these parameters. Were seeds from different species derived from the same genetic background or family line? These details must be included in the Materials and Methods section.

2.The sentence“This morphometric parameter is also important, as it shows the yield and strength of the selected rootstock” lacks sufficient justification. It is unclear how traits such as thousand-seed weight, plant weight, and root weight directly reflect yield performance. These are indicators of vegetative growth and do not necessarily correlate with reproductive output or fruit yield. The authors should provide evidence or references to support this claim. Additionally, the timing of these measurements is not specified—were these traits measured at the seedling stage, before or after grafting, or at maturity? Clarification is needed to understand their relevance and interpret the results accurately.

3.The experimental methods for measuring amino acids and IAA are insufficiently described. The target tissues or plant parts used for analysis were not specified. Furthermore, at which week post-grafting were the samples collected from grafted seedlings? These experimental details are essential and must be clearly provided in the Methods section.

4.The sentence “C. ficifolia samples are concentrated in the left segment of the graph, indicating significant differences between this species and the others in morphometric parameters” is not well-supported. Since only two samples of C. ficifolia were included, it is insufficient to conclude that there are significant differences between this species and the others based on such a limited sample size.

5.The rationale for selecting 1000-seed weight, plant height, and root length as the key parameters for rootstock selection is unclear. Why were these specific traits chosen? The authors should justify how these vegetative traits correlate with graft compatibility or scion performance.
6. In the biochemical analysis, the authors chose to measure IAA and amino acid content in leaves and stems, but not in roots. Since rootstock performance is closely related to root physiology, it would be more logical to include root tissues in the analysis. The authors should explain the rationale behind this tissue selection.
7. Lines 283–291 do not present actual results and therefore should not be included in the Results section. This content is more appropriate for the Discussion or Materials and Methods section.
8. The sentence in Line 312, “The highest content of many amino acids was characteristic of C. maxima, therefore this plant is the most preferable to use as a rootstock for cucumber,” is not sufficiently justified. A higher amino acid content alone does not necessarily make C. maxima the best rootstock. Rootstock performance should be evaluated based on multiple physiological and agronomic criteria.
9. In Figure 3, it is difficult to distinguish between stem and leaf samples in the central portion of the plot. There is no legend provided, making interpretation unclear. A proper legend and color/shape coding are essential for data clarity.

10.In Table 3, the data should be presented as mean ± standard deviation. Additionally, an ANOVA or appropriate statistical test should be conducted to assess whether the differences among groups are significant.

11.In Figure 4, what are the exact values for C. ficifolia and C. moschata stem samples? They appear to be close to zero. Please mark these values explicitly on the figure and re-check the underlying data for accuracy.

12.In Line 359, the sentence “The ligulate grafted plants (Fig. 5a) quickly grew together with the rootstock stem supports, and complete healing took one week” requires clarification. How was “complete healing” defined or assessed? Was this based on visual inspection, histological analysis, or another criterion?

13.In Lines 371–372, the term “grafted variant” is not appropriate.

14.Figures 6 and 7: Is the control group consistently defined? Earlier in the manuscript, it is stated that “C. sativus plants were also grafted onto themselves” as the control. Later, however, comparisons are made to “non-grafted control plants.” This inconsistency must be resolved. Each figure should clearly state which control is used (self-grafted or non-grafted) in the figure legend.

15.In Figure 7, only partial grafting data are shown. All grafted combinations, including those with different rootstocks, should be included in the analysis for comprehensive comparison.

---

## Round 0.2 · Minor Revisions

Please carefully check English and Figure captions and labeling.
Please check the figure quality.

**Language Note:** The review process has identified that the English language must be improved. PeerJ can provide language editing services - please contact us at [email protected] for pricing (be sure to provide your manuscript number and title). Alternatively, you should make your own arrangements to improve the language quality and provide details in your response letter. – PeerJ Staff

·

Basic reporting

The authors have made significant improvements to the manuscript. However, there are a few additional issues that need to be addressed. Firstly, the captions for the figures should be placed after the figures themselves. Additionally, while each figure is labeled with lowercase Roman letters, the captions currently use uppercase letters for these labels, especially the figures 5 onwards. The captions of all the figures should be revised for clarity to enhance readers' understanding.
All table captions should be written clearly to ensure readers can easily understand them.

Experimental design

No comment.

Validity of the findings

No comment.

Additional comments

No comments.

---

## Round 0.3 · accepted · Accept

The manuscript is much improved and can be accepted for publication.